# Exosomes from bone marrow mesenchymal stem cells protect melanocytes under vitiligo-related conditions through induction of NRF2/HO1 expression

Xuecheng Sun, Bo Huang, Gaobo Ruan, Aie Xu ⓘ*

Department of Dermatology, The Third People's Hospital of Hangzhou, Hangzhou, Zhejiang, P.R. China

* axu_hz3rd@sina.com

## Abstract

### Background

Vitiligo, a chronic autoimmune disease linked to excess oxidative stress, can be temporarily improved. Bone marrow mesenchymal stem cells (BMSCs)-derived exosomes (BMSCs-Exos) have recently emerged as a promising novel therapeutic means for vitiligo.

### Methods

Exosomes were isolated and characterized from BMSCs-conditioned medium. PIG3V cells and those transfected with *NRF2* siRNA or negative control were cultured under normal conditions or exposure to hydrogen peroxide ($H_2O_2$) to induce oxidative stress, with addition of BMSCs-conditioned medium, conditioned medium from BMSCs pretreated with GW4869 (referred to as BMSCs-GW4869), or BMSCs-Exos. Cell viability, apoptosis, and oxidative stress parameters, including cellular glutathione (GSH)/oxidized glutathione (GSSG) ratio, superoxide dismutase (SOD), reactive oxygen species (ROS), and malondialdehyde (MDA), were assessed. The expression of Ki67, NRF2, HO1, BAC, and Bcl-2 was measured.

### Results

BMSC-Exos significantly enhanced cell viability and reduced apoptosis and oxidative stress in $H_2O_2$-treated PIG3V cells. Simultaneously, BMSCs-Exos reversed $H_2O_2$-induced downregulation of Ki67, NRF2, HO1, and Bcl-2, and upregulation of BAX in PIG3V cells. Silencing *NRF2* by siRNA in PIG3V cells prior to $H_2O_2$ treatment abolished the protective effect of BMSCs-Exos and decreased the HO1 expression.

**Data availability statement:** All relevant data are within the manuscript and its Supporting Information files.

**Funding:** The Medicine and Hygiene Program of Zhejiang Province, Youth Innovation Project (No. 2023RC064). The funders had no role in study design, data collection and analysis, decision to publish, or preparation of the manuscript.

**Competing interests:** The authors have declared that no competing interests exist.

## Conclusions

BMSCs-Exos protect melanocytes from vitiliog-related oxidative stress by mitigating oxidative damage through induction of NRF2/HO1 expression.

---

## Introduction

Vitiligo is a chronic autoimmune disease causing depigmented patches and a reduced quality of life [1]. The global lifetime prevalence of vitiligo was estimated at 0.36% in adults and 0.24% in the child population [2]. The pathology of vitiligo is multimodal, involving excess oxidative stress, auto-reactive CD8$^+$ T-cells targeting melanocytes to release inflammatory mediators (such as IFN-γ), etc. [1]. For example, autoreactive cytotoxic CD8$^+$ T cells target melanocytes to promote the progression of vitiligo through production of inflammatory cytokines (such as IFN-γ), which in turn induces surrounding keratinocytes to secret chemokines that recruit cytotoxic CD8$^+$ T cells to the skin [3]. Current treatments that block IFN-γ signaling can temporarily ameliorate vitiligo symptoms, however, vitiligo often relapses after treatment is stopped [3], calling for more effective therapeutic strategies for vitiligo.

Mesenchymal stem cells (MSCs) are versatile cell populations typically obtained from adult bone marrow, adipose, umbilical cord and placenta. MSCs have a high capacity for extensive proliferation, multi-lineage differentiation, trophic function, homing/migration, immunosuppression, secretion of numerous growth factors, cytokines and exosomes, and low immunogenicity [4,5]. For the above characteristics, transplantation of MSCs is considered an promising treatment option for vitiligo [5]. For example, MSCs induce CD8$^+$ T cell apoptosis and suppress T cell proliferation via the NKG2D pathway, leading to repigmentation of vitiligo [6]. Capsaicin combined with MSCs promotes proliferation and alleviates mitochondrial dysfunction in human vitiligo melanocyte cell line PIG3V by downreguling the HSP70/TLR4/mTOR/FAK signaling axis [7]. MSCs also enhance melanocyte proliferation and inhibit oxidative stress-induced apoptosis by downregulating the phosphatase and tensin homolog pathway in vitiligo [8]. However, limited homing and survival of transplanted MSCs in target tissues [9,10], potential risks such as tumorigenicity during long-term maintenance [11], etc. remain significant challenges for clinical application of MSCs.

Exosomes are extracellular vesicles (40–160 nm in diameter) released by cells that contain nucleic acids (such as DNA, mRNA, and noncoding RNA), proteins (membrane, cytosolic, nuclear, and extracellular matrix proteins), lipids, and metabolites [12]. Exosomes mediate intercellular communication within and between tissues to participate in the regulation of immune responses, cardiovascular diseases, cancer progression, and many other diseases [12,13]. Compared with stem-cell based therapies, exosomes exhibit some key advantages. Exosomes are safe, non-tumorigenic, easy to handle, can be produced on a large scale, and do not require potentially toxic cryopreservative agents for maintenance, making off-the-shelf treatments feasible [14]. Therefore, exosomes are more suitable for cell-free regenerative therapy compared with their parental MSCs.

Despite growing interest, the effects of MSCs-derived exosomes on vitiligo and the underlying mechanisms remain unclear. In this study, we observed the protective role of bone marrow MSCs (BMSCs)-derived exosomes (BMSCs-Exos) in human melanocytes (PIG3V cells) exposed to $H_2O_2$-induced oxidative stress, an established *in vitro* model for vitiligo, and investigated the associated molecular mechanisms. This study may be beneficial to the potential clinical application of BMSCs-Exos in the treatment for vitiligo.

## Materials and methods

### BMSCs culturing and identification

BMSCs are characterized by the expression of surface markers CD29, CD44, CD105, CD73, and CD90, while lacking hematopoietic and myeloid markers such as CD34, CD45, and CD11b [15,16]. BMSCs were obtained from Cyagen Biosciences (Guangzhou, China) and were positive for CD29, CD44, and CD90, but negative for CD34, CD45, CD11b. BMSCs were cultured in Dulbecco's Modified Eagle Medium (Invitrogen, Waltham, MA, USA) supplemented with 10% fetal bovine serum (FBS) (Invitrogen) and 1% penicillin-streptomycin at 37°C with 5% $CO_2$. The culture medium was changed every 2−3 days. Osteogenic differentiation of BMSCs was assessed by culturing cells in osteogenic medium (WWL-G039, Fuheng Bio., Shanghai, China) containing dexamethasone, β-glycerophosphate. and ascorbic acid for 21 days, followed by Alizarin Red S (Fuheng Bio.) staining to detect the mineralization. Adipogenic differentiation was determined by culturing BMSCs in adipogenic medium (WWL-G040, Fuheng Bio.) containing dexamethasone, 3–isobutyl-1-methylxanthine, indomethacin, and insulin for 14 days, followed by Oil Red O (Fuheng Bio.) staining to visualize lipid droplets under a microscope. For chondrogenic differentiation, BMSCs were cultured in chondrogenic medium (WWL-G041, Fuheng Bio.) containing dexamethasone and TGF-β3 for 21 days, and Alcian Blue (Fuheng Bio.) staining was performed to detect glycosaminoglycans (GAGs).

### Isolation of exosomes

BMSCs-conditioned medium was collected and subjected to differential centrifugation, followed by ultracentrifugation at 100,000 g for 70 minutes. Pelleted exosomes were resuspended in phosphate buffered saline (PBS), aliquoted, and stored at −80°C. Exosomes were characterized by nanoparticle tracking analysis (NTA) for size distribution, transmission electron microscopy (TEM) for morphology, and Western blot analysis for the exosome markers CD9, CD63, and CD81.

### Western blot analysis

Cells were harvested and lysed in RIPA buffer (Thermo Fisher Scientific, Waltham, MA, USA) containing protease and phosphatase inhibitors (Roche, Germany). Protein concentrations of the cell lysates were measured using the BCA Protein Assay Kit (Thermo Fisher Scientific). Equal amounts (30–50 µg) of protein were separated by sodium dodecyl sulfate polyacrylamide gel electrophoresis and transferred on PVDF membranes (Millipore, Billerica, MA, USA). Membranes were blocked with 5% non-fat milk in Tris-buffered saline containing 0.1% Tween-20 (TBST) for 1 hour at room temperature, incubated overnight at 4°C with primary antibodies against CD9 (Cat# ab92726, Abcam, UK), CD63 (Cat# A19023, ABclonal, Woburn, MA, USA), Alix (Cat# ab186429, Abcam), Ki67 (Cat# HA721115, HUABIO, Hangzhou, China), NRF2 (Cat# A1244, ABclonal), HO1 (Cat# HA721854, HUABIO), BAX (Cat# 50599–2-Ig, Proteintech, Wuhan, China), BCL2 (Cat# 26593–1-AP, Proteintech), and GAPDH (Cat# 6004–1-Ig, proteintech), washed with TBST, and then incubated with HRP-conjugated secondary antibody (Cat# 7076, 7074, Cell Signaling Technology, Boston, MA, USA) for 1 hour at room temperature. After washing, protein bands were visualized using an enhanced chemiluminescence detection kit (Thermo Fisher Scientific). Densitometric analysis was subsequently performed using ImageJ software (NIH, USA) for semiquantitative evaluation of protein levels based on the optical density (OD) relative to the internal control GAPDH.

## Culturing and treatment of PIG3V cells

PIG3V cells (Qingqi (Shanghai) Biotech., Shanghai, China) were cultured in Keratinocyte-SFM medium (GIBCO, Jenks, OK, USA) supplemented with 5% FBS and 1% penicillin-streptomycin at 37°C, 5% $CO_2$. Keratinocyte-SFM is specifically formulated to support the growth and maintenance of human keratinocytes (such as PIG3V cells) under serum-free conditions, though it is sometimes supplemented with FBS and antibiotics (e.g., penicillin-streptomycin) to enhance cell support and prevent contamination. PIG3V cells were treated with 50% (v/v) conditioned medium from cultured BMSCs (referred to as BMSCs-medium), conditioned medium from BMSCs pretreated with GW4869 (an inhibitor of exosome release) at a final concentration of 5 μM (referred to as BMSCs-GW4869), or BMSCs-Exos at a final concentration of 20 μg protein/mL. Treatments were applied either under normal conditions or following hydrogen peroxide ($H_2O_2$) exposure to mimic vitiligo-related oxidative stress *in vitro*.

## Exosome uptake assay

PIG3V cells were seeded into 6-well plates at a density of $2 \times 10^5$ cells per well overnight. At 60–80% confluence, cells were incubated with BMSCs-Exos labeled with the lipophilic fluorescent dye DiI (Invitrogen) at a final concentration of 50 μg/mL at 37°C for 1, 4, 8, and 24 hours. The uptake of exosomes was evaluated by fluorescence microscopic observation.

## *NRF2* siRNA transfection

PIG3V cells at 60–80% confluence were transfected with *NRF2* siRNA (50–100 nM). *NRF2* siRNA sequences 5'-GGAUCCAGCAGUAAAGGATT-3' (forward) and 5'-UCCUUUACUGCUGGAUCCATT-3' (reverse) synthesized by Generay Shanghai Biotech. (Shanghai, China) were diluted in serum-free Opti-MEM medium (Thermo Fisher Scientific), and was then mixed with Lipofectamine 3000 reagent (Thermo Fisher Scientific). After 6–8 hours, medium was replaced with Keratinocyte-SFM medium supplemented with 10% FBS. Cells were cultured for 24–48 hours. Non-targeting siRNA sequences (Generay) were used as a negative control. The efficiency of transfection was assessed by Western blot analysis.

## Cell viability assay

Cell viability was evaluated using the Cell Counting Kit-8 (CCK-8, Dojindo, Japan) following the manufacturer's instruction. In brief, PIG3V cells were seeded ino 96-well plates and subjected to various treatment. Then, 10 μL CCK-8 reagent was added to each well, and the volume was adjusted to 100 μL with culture medium. After incubation for 1–2 hours at 37°C, the OD at 450 nm was detected using a microplate reader (BioTek, USA). The results were shown relative to untreated controls. Experiments were repeated five times.

## Assessment of apoptosis

Apoptosis was assessed using the Apoptosis Detection Kit (BD Biosciences, San Jose, CA, USA) following the manufacturer's instruction. In brief, PIG3V cells were harvested, washed, and resuspended in binding buffer at a density of $1 \times 10^6$ cells/mL, followed by incubation with 5 μL Annexin V-FITC and 5 μL of 7-Aminoactinomycin D (7-AAD) for 15 minutes at room temperature in the dark. The samples were then analyzed by flow cytometry (BD FACSCalibur, BD Biosciences). Annexin V-positive/7-AAD-negative cells were early apoptotic, whereas Annexin V-positive/7-AAD-positive cells were late apoptotic.

## Detection of reactive oxygen species (ROS)

PIG3V cells were harvested, resuspended in PBS at a density of $1 \times 10^6$ cells/mL, and incubated with 10 μM fluorescent dye 2',7'-dichlorofluorescin diacetate (DCFH-DA; Sigma-Aldrich) for 30 minutes at 37°C in the dark. After being washed

twice with PBS, intracellular ROS levels were measured by flow cytometry (BD FACSCalibur) using an excitation wavelength of 488 nm and emission wavelength of 530 nm. Data were analyzed using FlowJo software (TreeStar, USA), and the ROS contents were expressed as the percentage of DCFH-positive cells in the total cell population.

## Measurement of glutathione (GSH)/oxidized glutathione (GSSG), superoxide dismutase (SOD) and malondialdehyde (MDA)

Cells were harvested and homogenized in cold PBS, and the supernatant from the cell lysates was collected. Protein concentrations were determined using the BCA protein assay kit (Thermo Fisher Scientific). The levels of GSH and GSSG were measured using a GSH/GSSG Ratio Detection Assay Kit (BioVision, USA). The lysate supernatant was incubated with detection reagents, and the absorbance at 412 nm was recorded using a microplate reader to calculate the GSH/GSSG ratio. SOD activity was assessed by determining the reduction of WST-1 in the lysate supernatant at 450 nm using a SOD Assay Kit (Beyotime, Shanghai, China). MDA levels were quantified using a thiobarbituric acid-reactive substances assay kit (Beyotime) by detecting absorbance at 532 nm, and expressed as nmol/mg protein.

## Statistical analysis

Experiments were performed in triplicate or five independent repeats. Data are presented as the mean±standard deviations (SD) and were analyzed using GraphPad Prism (version 8.0.2). The Student's $t$-test was used for comparisons between two groups, and a $p < 0.05$ was considered statistically significant.

## Results

### Identification of BMSCs-Exos

BMSCs successfully underwent osteogenic, adipogenic, and chondrogenic differentiation, as evidenced by mineralization detected with Alizarin Red S staining (S1A Fig), lipid droplets visualized by Oil Red O staining (S1B Fig), and GAGs revealed by Alcian Blue staining (S1C Fig), confirming the multipotent potential of BMSCs.

Exosomes were successfully isolated from the conditioned medium of BMSCs. TEM revealed vesicles with a typical cup-shaped morphology (S1D Fig). NTA showed a mean particle diameter of approximately 100 nm (S1E Fig). Western blot analysis confirmed the presence of the exosomal markers CD9, CD63, and Alix in the isolated vesicles (S1F Fig). These findings are consistent with the characteristic features of exosomes.

### Time-dependent uptake of BMSCs-Exos by PIG3V cells

DiI-labeled BMSCs-Exos were added to PIG3V cell culture. Fluorescence microscopy revealed a time-dependent increase in the uptake of BMSCs-Exos, as evidenced by the progressive enhancement of red intracellular fluorescence (from DiI-labeled BMSCs-Exos) within PIG3V cells, whose nuclei were stained with DAPI (blue fluorescence). After 24 hours, most PIG3V cells had internalized BMSCs-Exos (Fig 1).

### BMSCs-Exos attenuated $H_2O_2$-induced injury in PIG3V cells

The CCK-8 assay showed that both BMSCs-conditioned medium and BMSCs-Exos significantly enhanced PIG3V cell viability under normal conditions ($p < 0.05$ and $p < 0.001$, respectively), with BMSCs-Exos exhibiting a stronger effect ($p < 0.001$). To mimic vitiligo-related oxidative stress *in vitro*, PIG3V cells were exposed to 100 μM $H_2O_2$ for 2 hours, which significantly inhibited cell viability compared with the control group ($p < 0.001$). BMSCs-conditioned medium partially restored PIG3V cell viability under $H_2O_2$ exposure ($p < 0.05$),whereas medium from BMSCs pretreated with exosome-release inhibitor GW4869 (BMSCs-GW4869) had no significant effect ($p > 0.05$). BMSCs-Exos more effectively improved PIG3V cell viability following $H_2O_2$ treatment ($p < 0.001$, Fig 2A,B).

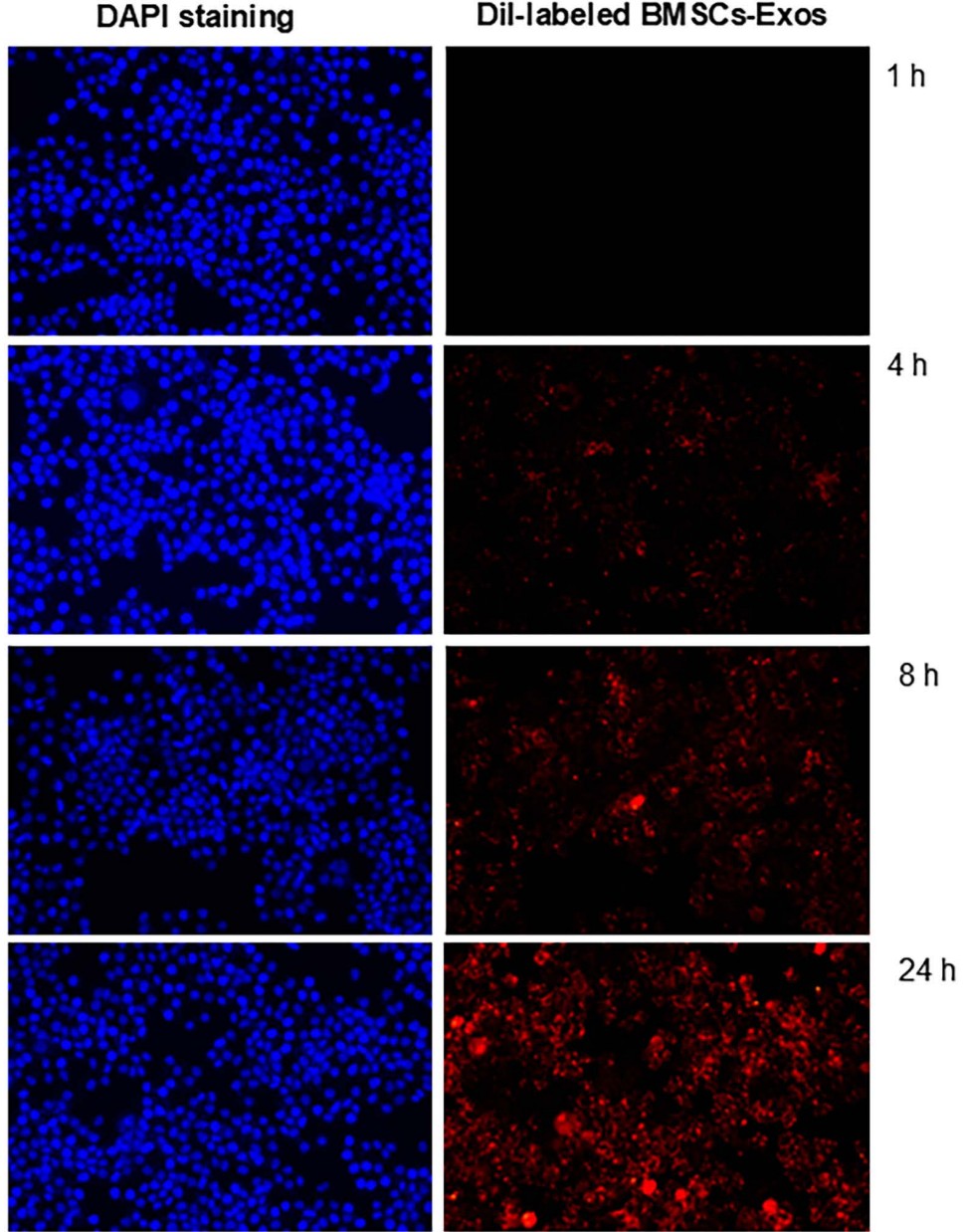

**Fig 1. Time-dependent uptake of BMSCs-Exos by PIG3V cells.** PIG3V cells at 60–80% confluence were incubated with DiI-labeled BMSCs-Exos for 1, 4, 8, and 24 hours at 37°C. Cells were collected and washed with PBS. The uptake of exosomes was evaluated by fluorescence microscopy (X100). Nuclei were then stained with 4′,6-diamidino-2-phenylindole (DAPI). Blue fluorescence (left) indicated DAPI-stained nuclei representing all PIG3V cells in a field. Red fluorescence (right) showed DiI-labeled BMSCs-Exos internalized by PIG3V cells.

Similarly, both BMSCs-conditioned medium and BMSCs-Exos significantly reduced apoptosis in PIG3V cells under normal and $H_2O_2$-treated conditions ($p < 0.001$), with BMSCs-Exos exhibiting a stronger anti-apoptotic effect (Fig 2C). Unexpectedly, BMSCs-GW4869 also significantly decreased apoptosis in PIG3V cells ($p < 0.001$, Fig 2C).

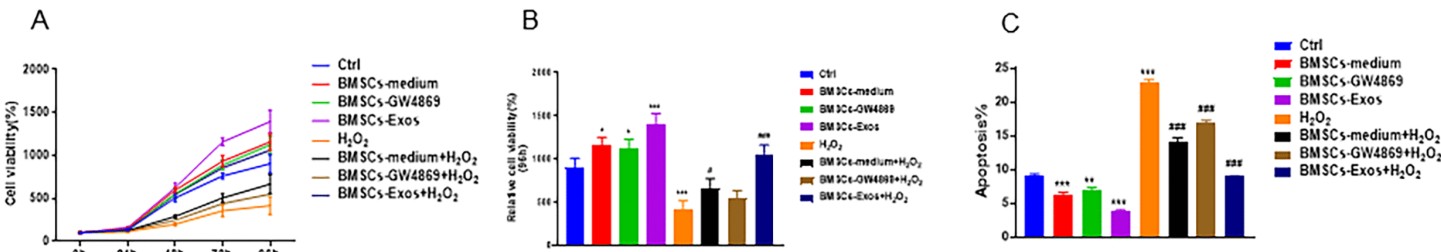

**Fig 2. BMSC-Exos attenuated H₂O₂-induced injury in PIG3V cells.** Untreated PIG3V cells and cells treated with BMSCs-conditioned medium, BMSCs-GW4869, or BMSCs-Exos were cultured under normal conditions or exposed to $H_2O_2$. The detailed description of treatment was shown in S1 Table. Cell viability was measured at 24, 48, 72, and 96 h (A), with detailed comparison at 96 h (B), and apoptosis was analyzed (C). Data were expressed as mean ± SD. $*p < 0.05$, $**p < 0.01$, $***p < 0.001$ vs. Ctrl; $\#p < 0.05$, $\#\#\#p < 0.001$ vs. $H_2O_2$.

### BMSCs-Exos reduced oxidative stress in H₂O₂-treated PIG3V cells

BMSCs-conditioned medium and BMSCs-Exos significantly increased the antioxidant markers GSH/GSSG ratio ($p < 0.01$) and SOD activity ($p < 0.001$) and decreased ROS levels ($p < 0.001$) in PIG3V cells. $H_2O_2$ exposure markedly decreased the GSH/GSSG ratio ($p < 0.01$) and SOD activity ($p < 0.001$) but increased ROS ($p < 0.001$) and MDA levels (indicative of lipid peroxidation, $p < 0.001$), indicating elevated oxidative stress in $H_2O_2$-treated PIG3V cells (Fig 3A–D). Both BMSCs-conditioned medium and BMSCs-Exos reversed the $H_2O_2$-induced decrease in the GSH/GSSG ratio ($p < 0.05$) and SOD activity ($p < 0.001$) and the increase in MDA ($p < 0.05$) and ROS ($p < 0.001$) levels, with BMSCs-Exos alleviating the oxidative stress more effectively (Fig 3A–D). BMSCs-GW4869 had no significant effect on oxidative stress parameters in $H_2O_2$-treated PIG3V cells (Fig 3A–D). These results highlight the antioxidative capacity of BMSCs-Exos in protecting PIG3V cells from $H_2O_2$-induced oxidative stress.

### Effect of BMSCs-Exos on the expression of proliferation-, oxidative stress-, and apoptosis-related key proteins in H₂O₂-treated PIG3V cells

Nuclear factor erythroid 2-like 2 (NRF2) functions as a transcription factor that induces the expression of its downstream target heme oxygenase-1 (HO1) to mitigate oxidative damage in various physiological and pathological processes by activating cellular protective antioxidant responses [17–19]. Therefore, we further examined the expression of NRF2 and HO1 in PIG3V cells in response to $H_2O_2$-treatment. In addition, the expression of proliferation-related Ki67 (an important indicator of cell proliferation), pro-apoptotic protein Bcl-2-associated X (BAX) [20], and anti-apoptotic protein Bcl-2 [21] were also evaluated.

Western blot analysis demonstrated that BMSCs-conditioned medium and BMSCs-Exos upregulated the expression of Ki67, NRF2, HO1, and Bcl-2 and downregulated BAX expression in PIG3V cells (Fig 4). $H_2O_2$ treatment markedly reduced Ki67, NRF2, HO1, and Bcl-2 expression and induced BAX expression, which was reversed by BMSCs-conditioned medium or BMSCs-Exos. Notably, the regulatory effects of BMSCs-Exos on the expression of Ki67, NRF2, HO1, BAX, and Bcl-2 were more pronounced than those of BMSCs-conditioned medium. BMSCs-GW4869 had no significant effects (Fig 4A, B).

### Silencing of *NRF2* attenuated the protective effects of BMSCs-Exos in H₂O₂-treated PIG3V cells

We observed that BMSCs-Exos restored NRF2 and HO1 expression in $H_2O_2$-Treated PIG3V cells. To further determine whether NRF2 mediates the protective effects of BMSCs-Exos under oxidative stress, *NRF2* expression was knocked down in PIG3V cells using siRNA. Transfection with *NRF2* siRNA significantly decreased cell viability ($p < 0.05$) and

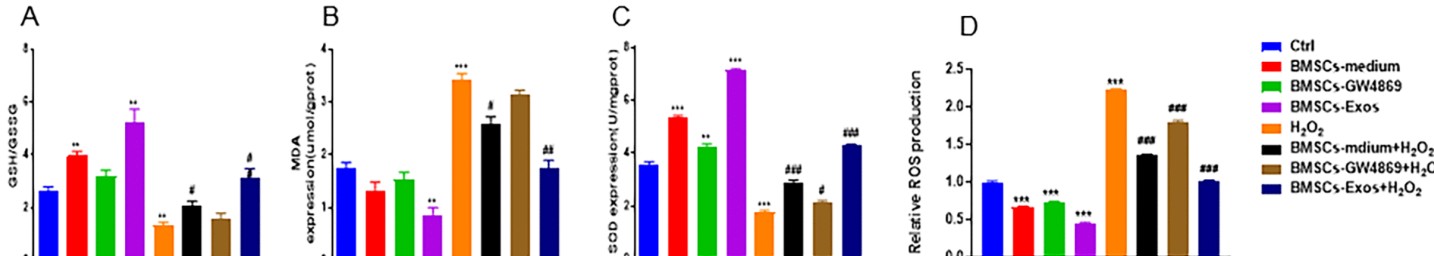

**Fig 3. Effect of BMSCs-Exos on the intracellular levels of GSH/GSSG ratio, SOD activity, ROS, and MDA in H$_2$O$_2$-treated PIG3V cells.** Untreated PIG3V cells and cells treated with BMSCs-conditioned medium, BMSCs-GW4869, or BMSCs-Exos were cultured under normal conditions or exposed to H$_2$O$_2$ treatment, as described in Fig 2 and S1 Table. Intracellular levels of GSH/GSSG ratio (A), SOD activity (B), ROS (C) and MDA (D) were measured. Data were expressed as mean ± SD. **p < 0.01, ***p < 0.001 vs. Ctrl; #p < 0.05, ##p < 0.01, ###p < 0.001 vs. H$_2$O$_2$.

increased apoptosis (p < 0.001) compared with NC siRNA-transfected cells. BMSCs-Exos significantly enhanced cell viability and reduced apoptosis in both NC siRNA- and *NRF2* siRNA-transfected cells (p < 0.001 and p < 0.01, respectively), but the effects of BMSCs-Exos on increasing cell viability (p < 0.05) and reducing apoptosis (p < 0.001) were markedly attenuated in *NRF2* siRNA-transfected cells (Fig 5A–C). Exposure to H$_2$O$_2$ significantly decreased cell viability (p < 0.01) and induced apoptosis (p < 0.001) in NC siRNA-transfected PIG3V cells, which was further exacerbated by *NRF2* knockdown (p < 0.05 and p < 0.001, respectively). BMSCs-Exos improved cell viability and reduced apoptosis in both NC siRNA- and *NRF2* siRNA- transfected PIG3V cells under H$_2$O$_2$ exposure (p < 0.05, p < 0.01, p < 0.001). The protective effects of BMSCs-Exos in H$_2$O$_2$-exposed PIG3V cells were significantly reversed by *NRF2* silencing (p < 0.05, p < 0.001; Fig 5A–C).

Furthermore, H$_2$O$_2$ exposure significantly reduced the GSH/GSSG ratio and SOD activity and increased MDA levels in PIG3V cells (p < 0.01, p < 0.001), which was further aggravated by *NRF2* knockdown (p < 0.05, p < 0.01). BMSCs-Exos markedly increased the GSH/GSSG ratio and SOD activity, and decreased MDA levels in H$_2$O$_2$-treated PIG3V cells (p < 0.01, p < 0.001), whereas *NRF2* silencing in PIG3V cells prior to H$_2$O$_2$ exposure significantly abolished these antioxidative effects (p < 0.01; Fig 5D–F).

### BMSCs-Exos regulated the expression of Ki67, BAX, and Bcl-2 through induction of NRF2 in H$_2$O$_2$-treated PIG3V cells

H$_2$O$_2$ treatment reduced the expression of Ki67, NRF2, HO1, and Bcl-2 and induced the expression of BAX in PIG3V cells. *NRF2* knockdown further exacerbated these changes. BMSCs-Exos upregulated the expression of Ki67, NRF2, HO1, and Bcl2, and downregulated BAX expression, whereas *NRF2* silencing partly reversed these changes in H$_2$O$_2$-treated PIG3V cells (Fig 6A, B).

### Discussion

While MSC transplantation is promising for the treatment of vitiligo, exosomes offer a more practical cell-free therapeutic alternative. Until now, however, the role and molecular mechanisms of MSCs-Exos in vitiligo are largely unknown. To date, only one study by Wang et al. reported that exosomes derived from human umbilical cord MSCs improved vitiligo by reducing CD8$^+$ T cells infiltration and promoting regulatory T cell expansions in skin, thereby exerting immunosuppressive effects, as well as by ameliorating oxidative stress-induced melanocyte injury through the delivery of miR-132-3p and miR-125b-5p targeting *Sirt1* and *Bak1*, respectively [22]. The role of BMSCs-exosomes in vitiligo and their underlying mechanisms have not been reported yet. In this study, we demonstrated that BMSCs-Exos attenuated H$_2$O$_2$-induced injury and reduced oxidative stress in human melanocyte cell line (PIG3V cells). Further study revealed that BMSCs-Exos

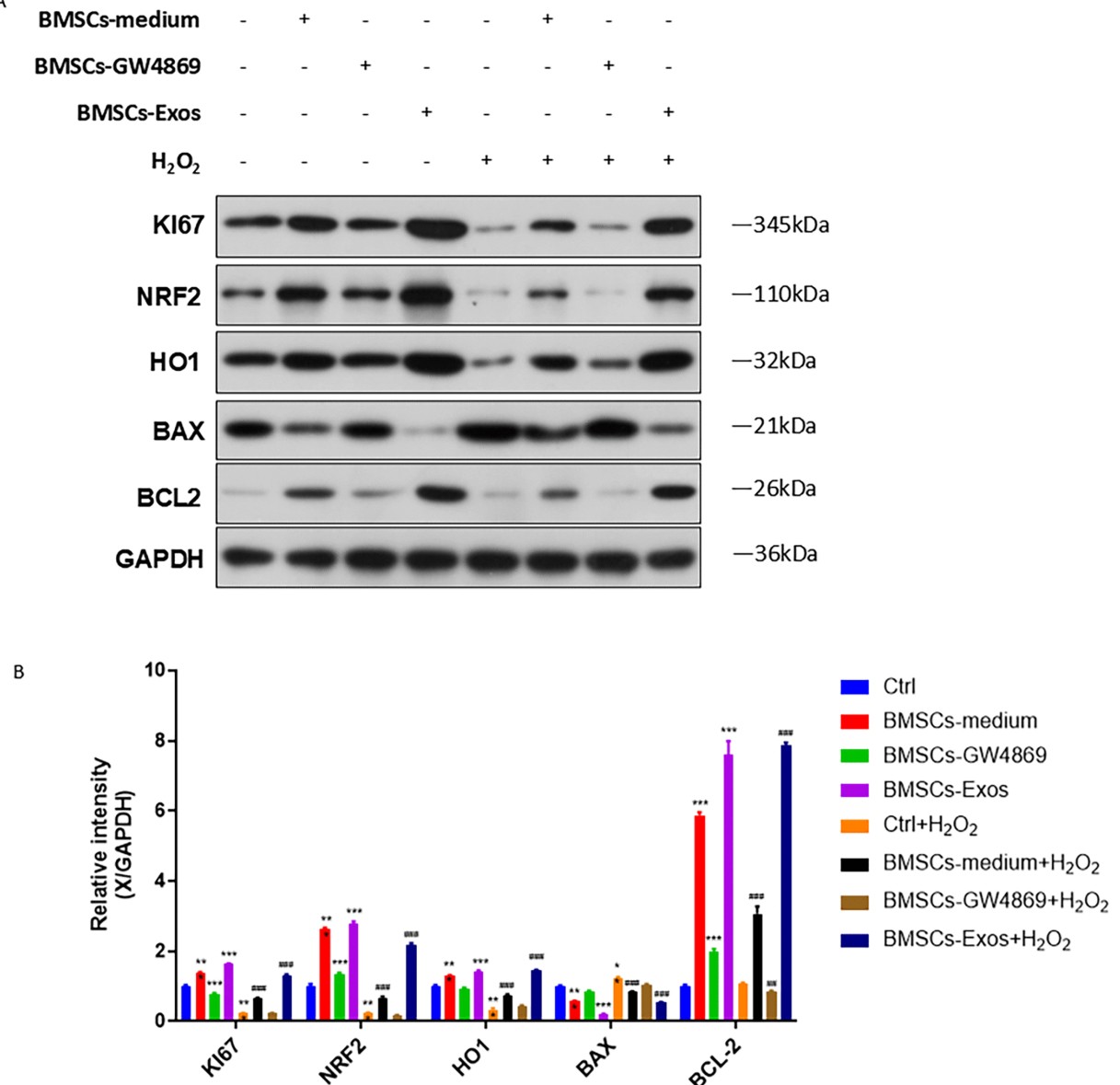

**Fig 4. Effect of BMSCs-Exos on the expression of Ki67, NRF2, HO1, BAX, and BCL2 in H$_2$O$_2$-treated PIG3V cells.** Untreated PIG3V cells and cells treated with BMSCs-conditioned medium, BMSCs-GW4869, or BMSCs-Exos were cultured under normal conditions or exposed to H$_2$O$_2$ treatment, as described in Fig 2 and S1 Table. Cells were harvested, and the protein expression of Ki67, NRF2, HO1, BAX, and BCL2 was determined by Western blot analysis (A) and quantified relative to GAPDH (B). Data were expressed as mean±SD. **$p < 0.01$, ***$p < 0.001$ vs. Ctrl; #$p < 0.05$, ##$p < 0.01$, ###$p < 0.001$ vs. H$_2$O$_2$.

functioned, at least in part, through the induction of NRF2/HO1. This study may be beneficial to clinical application of BMSCs-Exos in the treatment for vitiligo.

In this study, exosomes were derived from BMSCs-conditioned medium. We observed that BMSCs-conditioned medium significantly protected the viability of H$_2$O$_2$-treated PIG3V cells, reduced apoptosis, mitigated oxidative stress, and upregulated the expression of oxidative stress-related factors NRF2 and HO1. These protective effects might be

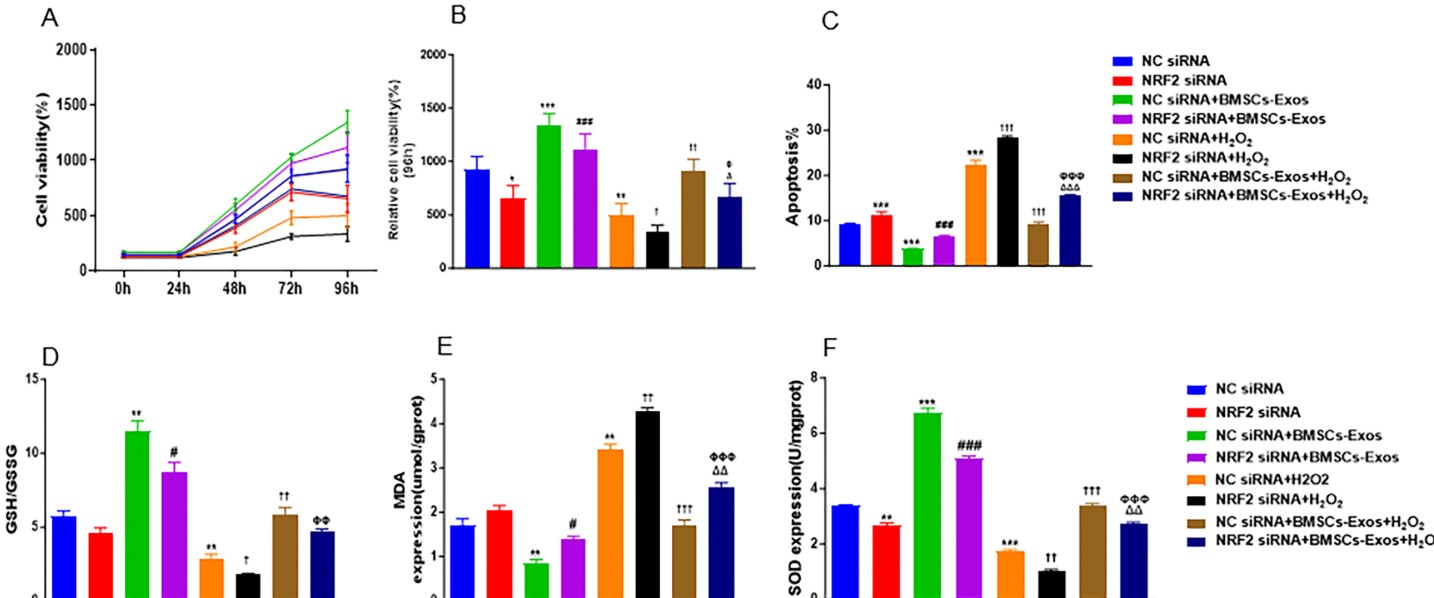

**Fig 5. Effect of *NRF2* silencing on the protective role of BMSCs-Exos in $H_2O_2$-treated PIG3V cells.** PIG3V cells transfected with negative control (NC) siRNA or *NRF2* siRNA were treated with or without BMSCs-Exos under normal conditions or $H_2O_2$ exposure. The detailed description of treatment was shown in S2 Table. Cell viability at 24, 48, 72, and 96 h (A), viability at 96 h (B), apoptosis (C), GSH/GSSG ratio (D), MDA content (E), and SOD activity (F) were assessed. Data were expressed as mean ± SD. *$p < 0.05$, ** $p < 0.01$, ***$p < 0.001$ vs. NC siRNA; ###$p < 0.001$ vs. NC siRNA+BMSCs-Exos; †$p < 0.05$, ††$p < 0.01$, †††$p < 0.001$ vs. NC siRNA + $H_2O_2$; Φ$p < 0.05$ vs. *NRF2* siRNA + $H_2O_2$; Δ$p < 0.05$, ΔΔΔ$p < 0.001$ vs. NC siRNA+BMSC-Exos+$H_2O_2$.

attributable to the exosomes present in the BMSCs-conditioned medium because the conditioned medium from BMSCs pretreated with GW4869, an inhibitor of exosome secretion, generally exhibit no obvious effects. Notably, purified BMSCs-Exos exerted stronger protective role in $H_2O_2$-treated PIG3V cells than BMSCs-conditioned medium, which might be due to the higher concentration of exosomes in BMSCs-Exos.

Although BMSCs-GW4869 mitigated the effect of BMSCs-conditioned medium, in some results it also significantly changed the viability and apoptosis of PIG3V cells. This probably could be due to inadequately suppression of release of exosomes from BMSCs into the conditioned medium at the relatively low concentration (5 μM) of GW4869 used in this study. In future work, exosomes will be quantified using NTA, exosome marker measurement, etc. to determine the inhibition efficacy of GW4869 in the release of exosomes. Moreover, more appropriate dosage of GW4869 in combination with silencing of exosomes activity-related genes will be employed to more effectively inhibit exosomes released from MSCs to further validate the role of BMSCs-Exos in vitiligo.

Oxidative stress-related NRF2/HO1 signaling mitigates oxidative damage in various physiological and pathological processes by activating cellular protective antioxidant responses [17–19]. In this study we found that $H_2O_2$ treatment significantly reduced the expression of NRF2 and HO1 in PIG3Vcells, which was effectively restored by BMSCs-Exos. Silencing the *NRF2* expression in PIG3Vcells attenuated the protective effects of BMSCs-Exos and reduced HO1 expression under $H_2O_2$ exposure. This result suggests that BMSCs-Exos exert protective effects at least in part through induction of the NRF2/HO1 signaling in PIG3Vcells.

We carried out a pioneering *in vitro* study to investigate the effect and underlying mechanisms of MSCs-derived exosomes in vitiligo. Compared with immortalized human melanocyte cell lines, primary human melanocytes more closely resemble *in vivo* physiological conditions, providing results with greater translational relevance and generalizability.

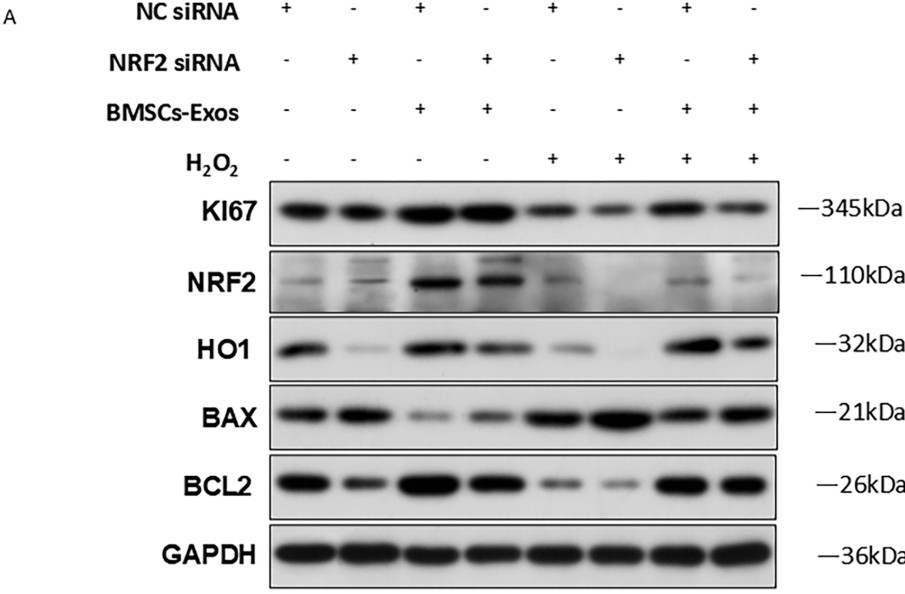

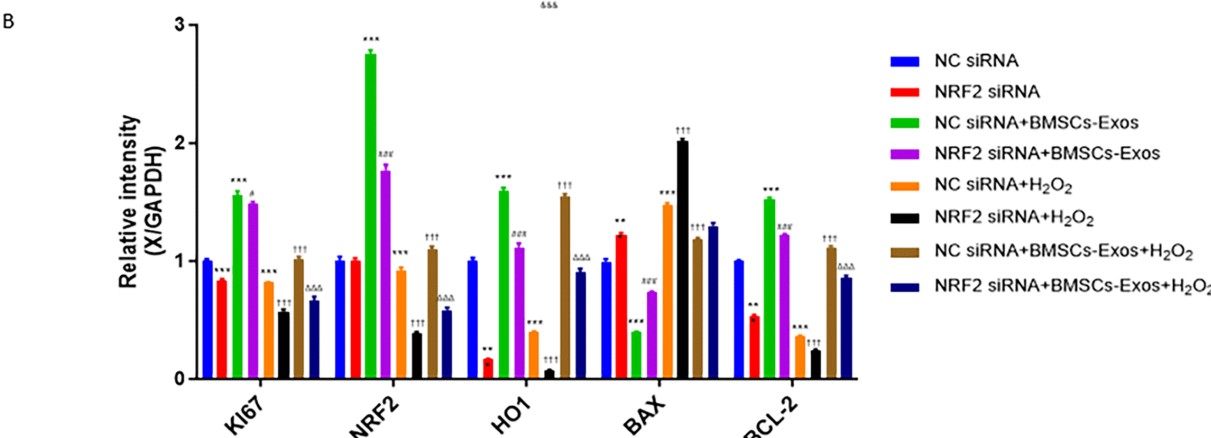

**Fig 6.** *NRF2* **silencing abolished the effect of BMSCs-Exos on the expression of Ki67, NRF2, HO1, BAX, and BCL2 in H₂O₂-treated PIG3V cells.** PIG3V cells transfected with NC sirNA or *NRF2* siRNA were treated with or without BMSCs-Exos under normal conditions or H₂O₂ exposure, as described in Fig 5 and S2 Table. The protein expression of Ki67, NRF2, HO1, BAX, and BCL2 was examined be Western Blot (A) and quantified relative to GAPDH (B). Data were expressed as mean±SD. **$p < 0.01$, ***$p < 0.001$ vs. NC siRNA; #$p < 0.05$, ###$p < 0.001$ vs. NC siRNA+BMSCs-Exos; †††$p < 0.001$ vs. NC siRNA+H₂O₂; △△△$p < 0.001$ vs. NC siRNA+BMSCs-Exos+H₂O₂.

However, the isolation and culture of primary human melanocytes require hospital ethical approval and considerable time. Therefore, in the current study, we used the human melanocyte cell line PIG3V. To enhance the translational value of the findings, primary human melanocytes will be isolated, cultured, and used in future studies to validate the results obtained from PIG3V cells. Moreover, a mouse model of vitiligo will be established and used to further validate these findings *in vivo.*

Until now, a study reported that adipose-derived MSC transplantation combined with ultraviolet B alleviated vitiligo in mice via induction of NRF2/HO1 expression [23]. However, no studies have yet investigated the role of NRF2/HO1 in the protective effects of exosomes in vitiligo. In this study, we demonstrated that BMSCs-Exos alleviated oxidative

stress-induced melanocyte damage associated with vitiligo through induction of NRF2/HO1, which has not been previously reported. This study did not identify the specific components of BMSCs-Exos (such as mRNA, noncoding RNAs, miRNAs, mRNAs, and proteins) that regulate NRF2/HO1 signaling to protect $H_2O_2$-treated PIG3V cells. In future studies, transcriptomic, proteomic and metabolomic analyses combined with *in vitro* and *in vivo* functional validation will be conducted. These investigations will focus on generating exosomes with altered expression of target molecules to identify upstream effectors (such as mRNAs, noncoding RNAs, miRNAs, and proteins) within BMSCs-Exos that activate the NRF2/HO1 signaling, thereby further elucidating the molecular mechanisms underlying the protective role of BMSCs-Exos in vitiligo.

## Conclusions

This study demonstrates that BMSCs-Exos protect melanocytes from vitiligo-related $H_2O_2$-induced injury by alleviating oxidative stress through induction of NRF2/HO1 expression. This study suggests that NRF2/HO1 signaling may serve as a potential therapeutic target for BMSCs-Exos-based treatment for vitiligo, which might be beneficial to the clinical application of BMSCs-Exos for vitiligo.

## Supporting information

**S1 Table. Experimental groups and treatments in PIG3V cells.**
(DOCX)

**S2 Table. Experimental groups and treatment in PIG3V cells.**
(DOCX)

**S1 Fig. Characteristics of BMSCs and BMSCs-Exos.** BMSCs were differentiated in (A) osteogenic medium for 21 days and stained with Alizarin Red S (X100), (B) adipogenic medium for 14 days and stained with Oil Red O (X100), and (C) chondrogenic medium for 21 days and stained with Alcian Blue (X100). Isolated BMSC-Exos were identified by TEM (X4000, D), NTA analysis (E), and Western blot analysis for exosomal markers CD9, CD63, and Alix, with GAPDH as internal control (F).
(TIF)

**S1 File. Raw data for line and column charts.**
(DOCX)

## Acknowledgments

We sincerely thank all members of our research team for their valuable contributions and technical support throughout this study. We also extend our gratitude to the institutional core facilities for providing access to essential equipment and resources..

## Author contributions

**Conceptualization:** Aie Xu.

**Data curation:** Xuecheng Sun, Gaobo Ruan.

**Formal analysis:** Xuecheng Sun, Gaobo Ruan.

**Funding acquisition:** Xuecheng Sun.

**Investigation:** Xuecheng Sun, Bo Huang, Gaobo Ruan.

**Supervision:** Aie Xu.

**Writing – original draft:** Xuecheng Sun.

**Writing – review & editing:** Aie Xu.

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
