## [Decision Letter · Decision Letter 0]

17 Sep 2025

Dear Dr. Xu,

Thank you for submitting your manuscript to PLOS ONE. After careful consideration, we feel that it has merit but does not fully meet PLOS ONE’s publication criteria as it currently stands. Therefore, we invite you to submit a revised version of the manuscript that addresses the points raised during the review process.

We look forward to receiving your revised manuscript.

Kind regards,

Vikash Chandra, PhD

Academic Editor

PLOS ONE

Journal Requirements:

“The Medicine and Hygiene Program of Zhejiang Province, Youth Innovation Project (No. 2023RC064)”

3. We note that your Data Availability Statement is currently as follows: [All relevant data are within the manuscript and its Supporting Information files.

” This work was supported by the Medicine and Hygiene Program of Zhejiang Province, Youth Innovation Project (No. 2023RC064).”

“The Medicine and Hygiene Program of Zhejiang Province, Youth Innovation Project (No. 2023RC064)”

Additional Editor Comments (if provided):

The manuscript "Exosomes from bone marrow mesenchymal stem cells protect melanocytes under vitiligo-related conditions through induction of NRF2/HO1 expression" is interesting; however, as reviewers stated, it needs to be revised according to their suggestions. Therefore, it is advisable to revise the manuscript in light of the reviewers' comments.

Reviewers' comments:

Reviewer's Responses to Questions

**Comments to the Author**

1. Is the manuscript technically sound, and do the data support the conclusions?

Reviewer #1: Yes

Reviewer #2: Partly

2. Has the statistical analysis been performed appropriately and rigorously?

Reviewer #1: Yes

Reviewer #2: Yes

3. Have the authors made all data underlying the findings in their manuscript fully available?

Reviewer #1: Yes

Reviewer #2: Yes

4. Is the manuscript presented in an intelligible fashion and written in standard English?

Reviewer #1: Yes

Reviewer #2: No

Reviewer #1: The manuscript is of good understanding towards the vitiligo and has potential to be carried in geenrating the therapeutic applications. Its well written and informed.

For improvement, manuscript may also provide details of the MSCs characterization as no detail on surface markers is given. Further, details about the role of each agent used in the vitiligo study and its therapeutic effect with the exosomes may be given.

Reviewer #2: Title: Exosomes from bone marrow mesenchymal stem cells protect melanocytes under vitiligo-related condition through induction of NRF2/HO1 expression

This is a well-designed and timely study that investigates a promising cell-free therapeutic strategy for vitiligo. The research question is clear and clinically relevant. The manuscript provides compelling in vitro evidence that exosomes derived from Bone Marrow Mesenchymal Stem Cells (BMSCs-Exos) can protect melanocytes from oxidative stress—a key driver of vitiligo pathogenesis—by activating the NRF2/HO1 antioxidant pathway. The experimental design is logical, and the data generally support the conclusions. The findings could have significant implications for developing novel treatments for vitiligo.

Strengths

1. The study addresses a significant unmet need in vitiligo treatment by exploring BMSCs-Exos as a safer, "off-the-shelf" alternative to whole MSC transplantation, mitigating risks like low engraftment survival and potential tumorigenicity.

2.The authors thoroughly evaluated the protective effects of BMSCs-Exos by assessing multiple endpoints: cell viability, apoptosis, and a panel of oxidative stress markers (GSH/GSSG, SOD, ROS, MDA). This multi-faceted approach strengthens the conclusions.

3. The use of conditioned medium from BMSCs treated with the exosome inhibitor GW4869 (BMSCs-GW4869) is a critical and well-considered control. It helps to attribute the observed effects specifically to the exosomes themselves rather than other soluble factors secreted by BMSCs.

4. The use of siRNA to knockdown *NRF2* expression is the strongest part of the manuscript. It effectively demonstrates that the protective effects of BMSCs-Exos are dependent on the NRF2/HO1 pathway, providing causal evidence for the proposed mechanism.

Shortcomings

1.The authors note that BMSCs-GW4869 still had a significant effect on apoptosis (Fig 2C), which they correctly attribute to potentially incomplete inhibition of exosome release. This is a limitation. The manuscript would be strengthened by including data (e.g., NTA or Western blot of markers) showing the actual reduction in exosome yield in the GW4869-conditioned medium to quantify the efficacy of inhibition.

2. The entire study is conducted in an in vitro cell line model (PIG3V). While this is a valid starting point, the conclusions about therapeutic potential for vitiligo would be significantly more powerful if supported by data from an in vivo animal model of vitiligo

3.The study identifies the NRF2/HO1 pathway as the mechanism but does not investigate what component of the BMSCs-Exos is responsible for activating this pathway. Is it specific miRNAs, proteins, lipids, or mRNAs? Identifying the active cargo would be a major advance and is a natural next step.

4.The findings are based on a single immortalized human melanocyte cell line (PIG3V). Repeating key experiments in primary human melanocytes would enhance the translational relevance and generalizability of the results.

5. The exosome inhibitor is consistently misspelled. It is GW4869, not "GW4849" (e.g., Abstract, Methods, Results, Figure legends).

6. Page 23, Line 3: "BMCs-Exos" should be "BMSCs-Exos".

7. Page 24, Line 1: "Bcl-X" is likely a typo and should be "BAX".

8. Page 25, Line 3: "NO-1" is a typo and should be "HO1".

9.The term "BMSCs-GW4849" is used in the text, but the figures (e.g., Fig 2, 3, 4) use the correct "BMSCs-GW4869". This must be harmonized.

10.The figure legends are very long and seemingly complicated, making them difficult to follow. The descriptions of each experimental group could be summarized in a table format within the legend or in a supplementary table for clarity.

11.The statistical annotation in Figure 5 is particularly complex and hard to decipher (e.g., `[5]`, `[55]`, `[&]`, `&&&`). Using more standard notations (e.g., asterisks and hashtags with clear definitions) would greatly improve readability.

12.In the Supplementary Figure legend (Page 35), it says "Fig 1B", "Fig 1C", etc. It should be "Fig S1B", "Fig S1C" for consistency.

13.While generally understandable, the manuscript would benefit from thorough proofreading by a native English speaker or a professional editing service to improve fluency. Some examples:

14.Page 11: "...causing white skin patches and reduced quality of life" -> "...causing depigmented patches on the skin and a reduced quality of life".

15.Page 12: "MSCs exhibit capacities of extensive proliferation..." -> "MSCs have a high capacity for extensive proliferation..."

16.Page 13: "Exosomes are safe without risk of tumorigenicity..." -> "Exosomes are safe, with no risk of tumorigenicity..."

what does this mean?

---

## [Author Response · Author response to Decision Letter 1]

28 Oct 2025

Dear Editor,

Thank you very much for reviewing our manuscript. We have carefully addressed all the issues raised by the editor and reviewers point by point, and have thoroughly revised the manuscript accordingly. These revisions are detailed in the uploaded files “Response to Reviewers,” “Revised Manuscript with Track Changes,” and “Manuscript.”. In addition, we have uploaded the raw data for the line and column charts (Figures 2-6) and the original, uncropped, and unadjusted Western blot images for Figures 4, 6, and S1 as Supporting Information files. All relevant data are included within the manuscript and its Supporting Information files. We confirm that this submission contains all raw data required to replicate the results of our study.

This work was supported by the Medicine and Hygiene Program of Zhejiang Province, Youth Innovation Project (No. 2023RC064). The funders had no role in study design, data collection and analysis, decision to publish, or preparation of the manuscript.

Should there be any questions, please do not hesitate to contact us.

Best regards

Aie Xu

3. We note that your Data Availability Statement is currently as follows: [All relevant data are within the manuscript and its Supporting Information files.

Response: Thank you for the suggestion. We have uploaded the raw data for the line and column charts (Figures 2-6) as a Supporting Information file.

Response: As suggested, we uploaded the original, uncropped, and unadjusted Western blot images for Figures 4, 6, and S1 as Supporting Information files. Thank you.

Response: The ORCID ID of the corresponding author has been linked to the submission account. Thank you for your kind reminder.

” This work was supported by the Medicine and Hygiene Program of Zhejiang Province, Youth Innovation Project (No. 2023RC064).”

“The Medicine and Hygiene Program of Zhejiang Province, Youth Innovation Project (No. 2023RC064)”

Response: Thank you for the suggestion. We have removed the funding description from the manuscript and added a funding statement in the Response to Reviewers as follows: This work was supported by the Medicine and Hygiene Program of Zhejiang Province, Youth Innovation Project (No. 2023RC064). The funders had no role in study design, data collection and analysis, decision to publish, or preparation of the manuscript.

Reviewers' comments:

Reviewer's Responses to Questions

Comments to the Author

5. Review Comments to the Author

Reviewer #1: The manuscript is of good understanding towards the vitiligo and has potential to be carried in geenrating the therapeutic applications. Its well written and informed.

For improvement, manuscript may also provide details of the MSCs characterization as no detail on surface markers is given.

Response: We appreciate the reviewer’s comment. BMSCs are characterized by the expression of surface markers CD105, CD44, CD29, CD73, and CD90, while lacking hematopoietic and myeloid markers such as CD34, CD45, and CD11b [15, 16]. In our study, BMSCs were obtained from Cyagen Biosciences (Guangzhou, China) and were positive for CD29, CD44, and CD90, but negative for CD34, CD45, CD11b. This information has been added to the subsection BMSCs culturing and identification (Page 5).

References:

15. Maličev E, Jazbec K. An Overview of Mesenchymal Stem Cell Heterogeneity and Concentration. Pharmaceuticals (Basel). 2024 Mar 7;17(3):350. doi: 10.3390/ph17030350. PMID: 38543135.

16. Gao Q, Wang L, Wang S, Huang B, Jing Y, Su J. Bone Marrow Mesenchymal Stromal Cells: Identification, Classification, and Differentiation. Front Cell Dev Biol. 2022 Jan 3;9:787118. doi: 10.3389/fcell.2021.787118. PMID: 35047499.

Further, details about the role of each agent used in the vitiligo study and its therapeutic effect with the exosomes may be given.

Response: We appreciate the reviewer’s valuable comment. We added a description of the role of Keratinocyte-SFM in vitiligo study in the subsection Culturing and treatment of PIG3V cells: “Keratinocyte-SFM is specifically formulated to support the growth and maintenance of human keratinocytes (such as PIG3V cells) under serum-free conditions, though it is sometimes supplemented with FBS and antibiotics (e.g., penicillin-streptomycin) to enhance cell support and prevent contamination” (Page 7-8).

The roles of GW4869 and H₂O₂ , which were applied to PIG3V cells, have already been addressed in the same subsection: “conditioned medium from BMSCs pretreated with GW4869 (an inhibitor of exosome release) at a final concentration of 5 µM (referred to as BMSCs-GW4869)”, and “Treatments were applied either under normal conditions or following hydrogen peroxide (H₂O₂) exposure to mimic vitiligo-related oxidative stress in vitro” (Page 8)

Reviewer #2: Title: Exosomes from bone marrow mesenchymal stem cells protect melanocytes under vitiligo-related condition through induction of NRF2/HO1 expression

This is a well-designed and timely study that investigates a promising cell-free therapeutic strategy for vitiligo. The research question is clear and clinically relevant. The manuscript provides compelling in vitro evidence that exosomes derived from Bone Marrow Mesenchymal Stem Cells (BMSCs-Exos) can protect melanocytes from oxidative stress—a key driver of vitiligo pathogenesis—by activating the NRF2/HO1 antioxidant pathway. The experimental design is logical, and the data generally support the conclusions. The findings could have significant implications for developing novel treatments for vitiligo.

Response: Thanks for these positive comments.

Strengths

1. The study addresses a significant unmet need in vitiligo treatment by exploring BMSCs-Exos as a safer, "off-the-shelf" alternative to whole MSC transplantation, mitigating risks like low engraftment survival and potential tumorigenicity.

2.The authors thoroughly evaluated the protective effects of BMSCs-Exos by assessing multiple endpoints: cell viability, apoptosis, and a panel of oxidative stress markers (GSH/GSSG, SOD, ROS, MDA). This multi-faceted approach strengthens the conclusions.

3. The use of conditioned medium from BMSCs treated with the exosome inhibitor GW4869 (BMSCs-GW4869) is a critical and well-considered control. It helps to attribute the observed effects specifically to the exosomes themselves rather than other soluble factors secreted by BMSCs.

4. The use of siRNA to knockdown *NRF2* expression is the strongest part of the manuscript. It effectively demonstrates that the protective effects of BMSCs-Exos are dependent on the NRF2/HO1 pathway, providing causal evidence for the proposed mechanism.

Response: Thanks for the above positive comments.

Shortcomings

1.The authors note that BMSCs-GW4869 still had a significant effect on apoptosis (Fig 2C), which they correctly attribute to potentially incomplete inhibition of exosome release. This is a limitation. The manuscript would be strengthened by including data (e.g., NTA or Western blot of markers) showing the actual reduction in exosome yield in the GW4869-conditioned medium to quantify the efficacy of inhibition.

Response: We appreciate the reviewer’s valuable comment. Our result showed that BMSCs-GW4869 mitigated the effect of BMSCs-conditioned medium in most experiments; however, in Fig 2C, both BMSCs-medium and BMSCs-GW4869 decreased apoptosis in H₂O₂-treated PIG3V cells. We speculate that this might be due to the used 5 µM dose of GW4849 being insufficient to fully suppress exosome release in BMSCs-conditioned medium. We acknowledge that quantifying exosomes via NTA, exosome marker measurement, etc. would help confirm the efficacy of inhibition. Unfortunately, due to missing samples and limited research funding, it was difficult for us to conduct these supplementary experiments. We have further discussed this in the Discussion section:

“Although BMSCs-GW4869 mitigated the effect of BMSCs-medium, in some results it also significantly affected the viability and apoptosis of PIG3V cells. This

probably could be due to inadequately suppression of release of exosomes from BMSCs into the conditioned medium at the relatively low concentration (5 μM) of GW4869 used in this study. In future studies, exosomes will be quantified using NTA, exosome marker measurement, etc. to determine the inhibition efficacy of GW4869 in the release of exosomes. Moreover, more appropriate dosage of GW4869 in combination with silencing of exosome activity-related genes will be employed to more effectively inhibit exosomes released from MSCs to further validate the role of BMSCs-Exos in vitiligo.” (The last paragraph of Page 17 and the first paragraph of Page 18)

2. The entire study is conducted in an in vitro cell line model (PIG3V). While this is a valid starting point, the conclusions about therapeutic potential for vitiligo would be significantly more powerful if supported by data from an in vivo animal model of vitiligo

Response: We appreciate the reviewer’s constructive comment. This study is an in vitro investigation, demonstrating that BMSCs-Exos protect melanocytes from vitiliog-related H₂O₂-induced oxidative stress through induction of NRF2/HO1 expression. We agree that the conclusion would be further strengthened by in vivo experiments. Nevertheless, constrained by time and research funding, these experiments could not be completed within the current study period. In future work, a mouse model of vitiligo will be established and used to validate these findings in vivo. This has been addressed in the Discussion section. (Page 19)

3.The study identifies the NRF2/HO1 pathway as the mechanism but does not investigate what component of the BMSCs-Exos is responsible for activating this pathway. Is it specific miRNAs, proteins, lipids, or mRNAs? Identifying the active cargo would be a major advance and is a natural next step.

Response: We appreciate this valuable suggestion. Identifying the specific components of BMSCs-Exos (such as mRNA, noncoding RNAs, miRNAs, mRNAs, and proteins) that regulate NRF2/HO1 signaling to protect H2O2-treated PIG3V cells would really be a significant advancement. To further elucidate the related molecular mechanisms, transcriptomic, proteomic and metabolomic analyses will be conducted to screen differentially expressed miRNAs, mRNAs, proteins, etc. Subsequently, exosomes carrying target mRNA/shRNA, miRNA minic/inhibitor, etc. will be generated through genetic manipulation of BMSCs and used for functional validation in vitro and in vivo. However, these experiments, which constitute a series of studies, are expected to take at least one year and could not be completed within the current study frame. This has been discussed in the Discussion section:

“This study did not identify the specific components of BMSCs-Exos (such as mRNA, noncoding RNAs, miRNAs, mRNAs, and proteins) that regulate NRF2/HO1 signaling to protect H2O2-treated PIG3V cells. In future studies, transcriptomic, proteomic and metabolomic analyses combined with in vitro and in vivo functional validation will be conducted. These investigations will focus on generating exosomes with altered expression of target molecules to identify upstream effectors (such as mRNAs, noncoding RNAs, miRNAs, and proteins) within BMSCs-Exos that activate the NRF2/HO1 signaling, thereby further elucidating the molecular mechanisms underlying the protective role of BMSCs-Exos in vitiligo.” (Page 19).

4.The findings are based on a single immortalized human melanocyte cell line (PIG3V). Repeating key experiments in primary human melanocytes would enhance the translational relevance and generalizability of the results.

Response: Thank you for the valuable suggestion. As mentioned in Introduction, the effect and underlying mechanisms of MSCs-derived exosomes in vitiligo remain unclear. Therefore, we carried out a pioneering in vitro study demonstrating that BMSCs-Exos protect melanocytes exposed to vitiliog-related H₂O₂-induced oxidative stress by induction of NRF2/HO1 expression. Compared with immortalized human melanocyte cell lines, primary human melanocytes more closely resemble in vivo physiological conditions, providing results with greater translational relevance and and

---

## [Decision Letter · Decision Letter 1]

21 Nov 2025

Exosomes from bone marrow mesenchymal stem cells protect melanocytes under vitiligo-related conditions through induction of NRF2/HO1 expression

PONE-D-25-38005R1

Dear Dr. Xu,

We’re pleased to inform you that your manuscript has been judged scientifically suitable for publication and will be formally accepted for publication once it meets all outstanding technical requirements.

Kind regards,

Vikash Chandra, PhD

Academic Editor

PLOS ONE

Reviewers' comments:

Reviewer's Responses to Questions

**Comments to the Author**

Reviewer #1: All comments have been addressed

Reviewer #2: All comments have been addressed

2. Is the manuscript technically sound, and do the data support the conclusions?

Reviewer #1: Yes

Reviewer #2: Yes

3. Has the statistical analysis been performed appropriately and rigorously?

Reviewer #1: I Don't Know

Reviewer #2: Yes

4. Have the authors made all data underlying the findings in their manuscript fully available?

Reviewer #1: Yes

Reviewer #2: Yes

5. Is the manuscript presented in an intelligible fashion and written in standard English?

Reviewer #1: Yes

Reviewer #2: Yes

Reviewer #1: The manuscript is of importance as it focuses on an important aspect of the biology, vitiligo.

May be accepted as I feel it is ok

Reviewer #2: The authors' revisions are comprehensive and have successfully addressed all concerns raised by both reviewers. The manuscript is significantly improved in terms of scientific rigor, clarity, and presentation. This work makes a valuable contribution to the field by establishing a novel protective role and mechanism for BMSCs-Exos in vitiligo, providing a solid foundation for future research.

---

## [Editor Report · Acceptance letter]

PONE-D-25-38005R1

PLOS ONE

Dear Dr. Xu,

I'm pleased to inform you that your manuscript has been deemed suitable for publication in PLOS ONE. Congratulations! Your manuscript is now being handed over to our production team.

Kind regards,

on behalf of

Dr. Vikash Chandra

Academic Editor

PLOS ONE